# Quitting Smoking before and after Pregnancy: Study Methods and Baseline Data from a Prospective Cohort Study

**DOI:** 10.3390/ijerph191610170

**Published:** 2022-08-17

**Authors:** Erica Cruvinel, Kimber P. Richter, Kathryn I. Pollak, Edward Ellerbeck, Nicole L. Nollen, Byron Gajewski, Zoe Sullivan-Blum, Chuanwu Zhang, Elena Shergina, Taneisha S. Scheuermann

**Affiliations:** 1Department of Population Health, University of Kansas Medical Center, Kansas City, KS 66160, USA; 2Department of Population Health Sciences, and Cancer Prevention and Control Program, Duke Cancer Institute, Duke University School of Medicine, Durham, NC 27705, USA; 3Department of Biostatistics & Data Science, University of Kansas Medical Center, Kansas City, KS 66160, USA

**Keywords:** longitudinal studies, smoking, pregnant women, postpartum period, methods

## Abstract

Smoking during pregnancy and postpartum remains an important public health problem. No known prior study has prospectively examined mutual changes in risk factors and women’s smoking trajectory across pregnancy and postpartum. The objective of this study was to report methods used to implement a prospective cohort (Msgs4Moms), present participant baseline characteristics, and compare our sample characteristics to pregnant women from national birth record data. The cohort study was designed to investigate smoking patterns, variables related to tobacco use and abstinence, and tobacco treatment quality across pregnancy through 1-year postpartum. Current smokers or recent quitters were recruited from obstetrics clinics. Analyses included Chi-square and independent sample t-tests using Cohen’s *d*. A total of 62 participants (41 smokers and 21 quitters) were enrolled. Participants were Black (45.2%), White (35.5%), and multiracial (19.3%); 46.8% had post-secondary education; and most were Medicaid-insured (64.5%). Compared with quitters, fewer smokers were employed (65.9 vs 90.5%, Cohen’s *d* = 0.88) and more reported financial strain (61.1% vs 28.6%; Cohen’s *d* = 0.75). Women who continue to smoke during pregnancy cope with multiple social determinants of health. Longitudinal data from this cohort provide intensive data to identify treatment gaps, critical time points, and potential psychosocial variables warranting intervention.

## 1. Introduction

Smoking during pregnancy is a key modifiable risk factor for poor maternal and infant health outcomes. Smoking is linked to preterm birth, stillbirth, neonatal mortality, miscarriage, and fetal growth restriction [1,2,3]. Furthermore, smoking during pregnancy is associated with long-term consequences for the child in terms of growth, delayed development, and weight problems [4,5,6,7]. Children of smokers have a higher incidence of childhood asthma, behavioral disorders, and poor academic performance in school [8,9]. Despite these problems, in the United States (US), one in fourteen women smoke during pregnancy [10]. Some women quit tobacco use when they learn about their pregnancy, but most return to smoking following birth [11,12,13,14].

Cross-sectional and longitudinal studies have identified important variables related to tobacco use and relapse both during pregnancy and postpartum such as socioeconomic status, depression, and having a partner who smokes [11,12,14,15,16]. In the United Kingdom (UK), Munafò and investigators found that a reduction in depressive symptoms during the course of pregnancy to the immediate postnatal period was associated with smoking cessation [15]. A more recent cohort in the UK showed that while quit attempts increased late in pregnancy, the intention to quit in the next 30 days decreased in the same period of time. Further, both quit attempts and intention to quit decreased 3 months after delivery [17]. In the US, a prospective longitudinal cohort tracking the number of cigarettes used per month from preconception to 2-months postpartum showed that most women cut down their smoking between the third and fourth month of pregnancy [18]. Together, these studies show that women’s smoking changes during the peripartum period and that psychosocial variables are important determinants of smoking. Prior pregnancy and postpartum cohorts were unable to determine the precise timing of relapse among quitters since smoking status was assessed in a single evaluation at 3-months postpartum [17] or at only a few timepoints during the first year postpartum [12,15]. Even long cohorts that followed mothers up to 6 years post childbirth, assessed smoking status in only six timepoints during the entire 6-year period [11]. Another limitation of prior cohort studies is that they assessed few potential psychosocial variables related to tobacco use that can be informative for intervention development.

Similarly, few studies have explored whether and how smoking cessation treatment is provided during health care visits. Several studies have suggested that health care providers routinely ask about tobacco use in the initial prenatal visits but do not routinely provide other elements of guidelines-based tobacco treatment [19,20,21]. In the UK, almost 40% of women reported no discussion of quitting with a health care provider across the entire pregnancy despite the fact that approximately half of the women were interested in receiving support to stop smoking [22]. Many of these prior studies have relied on retrospectively reported tobacco use data for preconception and pregnancy [11,12] or did not assess concurrent changes in psychosocial variables related to tobacco use that can be informative for intervention development [11,12,14,17,18,22,23]. Importantly, no studies have prospectively examined associations between women’s perceptions of the quality of smoking cessation treatment during pregnancy and postpartum and any resultant changes in smoking status.

In order to improve our understanding of proximal influences on women’s peripartum smoking patterns, a prospective cohort study was conducted, Msgs4Moms, enrolling women during pregnancy and following them through 1-year postpartum. This longitudinal cohort study leveraged text messaging, a widely used, easily accessed, and low-cost communication method, as a novel approach for assessing smoking at weekly intervals. Text messages were also used to track when participants visited their health care provider and to conduct brief surveys on the quality of the tobacco treatment they received. The study also longitudinally assessed a wide range of potential variables related to tobacco use during pregnancy and the first year postpartum using five online surveys delivered at intervals throughout pregnancy and postpartum. Specifically, our surveys assessed psychological variables (e.g., depression, anxiety, coping, and stress) [24,25], financial and food insecurity, social environmental influences [24,26], substance use [16,27,28], and other relevant health behaviors. To our knowledge, the present cohort is the first study to prospectively track smoking behavior changes using weekly remote assessments of cigarette use from pregnancy to 1-year postpartum. It is also the first cohort to proximally assess the quality of tobacco treatment delivered across health care visits.

This cohort study was developed to (1) identify detailed patterns of smoking and quitting during pregnancy and postpartum, (2) describe the sociodemographic and psychosocial variables related to women’s peripartum smoking patterns, and (3) describe women’s perspectives on the quality of tobacco treatment received during pregnancy, postpartum, and pediatric visits.

The aim of this paper was to describe the novel methods used to implement the cohort study, collect intensive smoking status data over the course of pregnancy and the first year postpartum, and we present the baseline sociodemographic and tobacco-related characteristics of cohort participants. In addition, we compare demographic and smoking characteristics of the recruited sample with pregnant women from national birth record data to examine the representativeness of this cohort.

## 2. Materials and Methods

### 2.1. Design

The study was a prospective cohort design. Women were followed from enrollment during pregnancy to 1-year postpartum. The longitudinal follow-up included three types of assessments (see Figure 1): (1) a baseline and four follow-up email surveys assessing tobacco-related risk factors, (2) six surveys sent via text message with a survey link assessing the quality of tobacco treatment that participants received during prenatal or postpartum visits, and (3) weekly text messages assessing past 7-day tobacco use and craving. STROBE guidance was used for reporting [29]. Institutional Review Board approval was received from the University of Kansas Medical Center Institutional Review Board.

### 2.2. Sample Size

The required sample size of 48 participants was based on 80% power to detect a small to medium (0.4) correlation between the quality of tobacco treatment and length of abstinence at *p* < 0.05 using a two-tailed test. Data from the National Health Interview Survey indicate a medium association between physician advice to quit and cessation [30]. Recruiting 60 participants with an expected retention rate of 80% would result in at least 48 women completing the study.

### 2.3. Participants

A cohort of 62 pregnant women was recruited. Eligibility criteria included having smoked at least 100 cigarettes in their lifetime, age 18 years of age or older, being current smokers or having smoked anytime in the 6 months prior to pregnancy (recent quitters), English-speaking, access to a cellphone, and being willing to receive and send text messages. We initially excluded women greater than 28 weeks gestational age; however, this exclusion criterion was removed to improve our ability to recruit participants.

### 2.4. Study Procedures

#### 2.4.1. Recruitment

Women were recruited between March 2019 and January 2020 from two Kansas and Missouri metro areas. The full sample was recruited from the University of Kansas Medical Center (KUMC) and from external health care clinics. Participants were recruited while attending prenatal appointments at five obstetric and family practice clinics and a prenatal educational program in partnership with a local health department. At external clinic recruitment sites, potentially eligible women completed a consent to contact form that included prescreening questions regarding smoking history. Clinic staff emailed or faxed the authorization to contact potential patients to the study team. At KUMC, study staff used the electronic medical record to identify potential participants and met with the patients during their clinic visit. Study staff provided a brief overview of the study and asked interested participants to complete the consent to contact form. In addition, the investigators explored using Facebook ads as a recruitment tool for a 2-week period but were unable to reach any of the potential participants who provided contact information.

Women willing to participate in the cohort study were contacted by study staff to complete screening over the phone. Eligible women verbally consented to participate in the cohort study and study staff assessed comprehension of the study requirements and potential risks. Eligible participants were emailed a copy of the consent form and a baseline survey. Women who completed the baseline survey within 2 weeks were enrolled in the study.

#### 2.4.2. Emailed Survey Distribution

Participants were sent emailed invitations to complete a baseline survey and four follow-up surveys at the following time-points: third trimester, 1-month postpartum, 6-months postpartum, and 1-year postpartum. The surveys assessed socio-demographic and pregnancy characteristics, tobacco and other substance use, and psychosocial variables.

Retention efforts for the study included reminder emails for survey completion and follow-up calls from study staff. After the initial email invitation for each survey, survey reminders with the link to the survey were sent every 2 days for five occurrences. After this, study staff called participants until they either filled out the questionnaire or their survey 60-day window expired. Study staff also sent mailed paper surveys if unable to contact by phone or email.

#### 2.4.3. Weekly Tobacco Use Text Survey Distribution

Participants were sent weekly brief text messages from enrollment through to 1-year postpartum. These weekly text surveys assessed cigarette smoking and craving over the past 7 days. Participants were asked about recent (past 7 days) and upcoming health care provider appointments.

#### 2.4.4. Tobacco Treatment Quality Questionnaire Distribution

The six surveys assessing provider tobacco treatment quality were administered following two prenatal, the postpartum, and three well-baby visits. These surveys were sent via a text message survey link. To reduce recall errors, the tobacco treatment quality surveys were scheduled using appointment dates from the weekly text message questions. Surveys about tobacco treatment quality were sent within an hour of responding to the text message for recent appointments or scheduled for the date of upcoming appointments. The survey was scheduled to be resent every 24 h for five occurrences, with a 7-day window for completion.

All study surveys were stored and distributed using REDCap. REDCap is a secure web-based application for developing surveys, capturing data, and managing data for research studies [31,32]. Text assessments were delivered via REDCap using the Twilio API (a text service provider) to deliver the text message inquiries and responses, which were stored in REDCap. All surveys in this study used primarily automated delivery. Each participant received a unique identifier that identified all sent and received messages and surveys.

#### 2.4.5. Incentives

Participants received a USD 30 incentive at baseline, a USD 25 incentive for each follow-up survey (3rd trimester,1-, 6-, and 12-months postpartum) and an additional USD 10 if they completed the surveys within 1 week. They received also USD 5 for completing each survey following a health care visit (PEIs) and a bonus of USD 5 if they completed this survey within 5 days. Participants received a USD 1 incentive for each weekly text message survey with a bonus payment of USD 1 at the end of the month if they completed all the surveys for any given month. Participants could receive up to a maximum total of USD 330 for participating in this study. Incentives were electronically deposited to a cash card that could be used as a debit card.

### 2.5. Survey Measures

Survey time points and measures are presented in Table 1. Participant demographics and pregnancy-related characteristics including age, education, race, ethnicity [33], household income, employment status, transitions in housing, parity, and due date were collected at baseline.

#### 2.5.1. Tobacco Outcome Measures

The 7-day point prevalence smoking status was assessed each week, including whether participants smoked, cigarettes per day, number of days smoked, and craving. Use of other tobacco products, e-cigarettes, and quit attempts were also assessed in each emailed survey.

#### 2.5.2. Variables Related to Tobacco Use

The cohort collected measures identified in prior studies as factors related to abstinence/relapse in this population [34]. These include smoking history and related characteristics (e.g., nicotine dependence) [35], smokers in the social network, motivation and confidence to quit [36], and psychological variables (e.g., stressors, depression, and anxiety) [37,38].

#### 2.5.3. Quality of Tobacco Treatment

This study used an adapted version of the Patient Exit Interview (PEI) [39] to measure the degree to which participants’ health care providers delivered the evidence-based 5 “A”s of tobacco treatment. The PEI yields a single score that represents the quality of tobacco treatment delivered by providers. The PEI has good validity [40] and can be administered in 3–5 min.

### 2.6. Data Analysis

Descriptive statistics (means and standard deviations for continuous variables and totals/percentages for categorical variables) were used to summarize baseline demographic and tobacco-related measures by smoking status. We calculated Cohen’s *d* to indicate effect size [41]. Cohen’s *d* is primarily used to report intervention effect sizes; however, effect sizes may be useful for interpreting the magnitude of differences between groups in observational studies [42,43]. Cohen’s *d* values are: <0.2 = negligible effect, ≥0.2 to <0.5 = small effect, ≥0.5 to <0.8 = medium effect and ≥0.8 = large. For continuous variables d = (mean1-mean2)/SD, for binary data d = ln (OR12) × sqrt(3)/pi, where ln is the natural log, OR12 is the odds ratio of group 1 versus group 2, and pi is approximately 3.14.

We compared demographics of the cohort sample to the 2018 Natality data from the Center for Disease Control and Prevention (CDC). The US Natality dataset reports statistics from birth certificates within the United States. Data are available by a variety of demographic characteristics, such as state and county of residence, mother’s race, and age, as well as maternal health risk factors, such as tobacco use during pregnancy [44]. Analyses were carried out in SAS version 9.4.23 (SAS Institute, Cary, NC, USA).

## 3. Results

A total of 117 women completed consent to contact forms at the recruitment sites. In total, 45 (38.5%) women were not eligible to participate in the study, resulting in an eligibility rate of 61.5%. Most participants were excluded for being unable to reach or consent, for not being interested in the research, or for being a non-smoker. Among 72 eligible and consented participants, 62 completed the baseline questionnaire within 2 weeks and were enrolled in the study (see Figure 2). The cohort was comprised of 41 women who were currently smoking at baseline and 21 who were recent quitters.

Table 2 presents demographic and psychosocial characteristics of the 62 women enrolled in the study. Almost half of participants were Black or African American (45.2%), had some college education or a college degree (46.8%), and most were Medicaid-insured (64.5%). At baseline, participants were on average 18.9 weeks pregnant. Most women reported financial strain, such as difficulty paying bills (50.0%), worrying about money to pay rent (50%), or not having enough money to pay for meals (67.7%).

Current smokers (n = 41) and recent quitters (n = 21) differed across a range of characteristics (Table 2). A higher percentage of White participants were former smokers compared with current smokers (52.4% vs 26.8%) with a medium effect size of 0.61. While 24.4% of current smokers had less than a high school diploma, only 4.8% of former smokers had the same level of education (Cohen’s *d* = 1.03). More participants were employed among former smokers than among current smokers (90.5% vs 65.9%), with a large effect size of 0.88. Six in ten current smokers reported not having enough money to pay bills and rent (61.1% vs 28.6%) with a large effect size of 0.75.

Six months before becoming pregnant, most participants were smoking every day (87.1%) (see Table 3). A higher percentage of current smokers smoked daily prior to pregnancy (92.7%) compared with former smokers (76.2%), with a medium effect size of 0.76. The use of other tobacco products during pregnancy was also higher among current smokers (24.4% vs 9.5%, Cohen’s *d* = 0.62). More than a quarter of women used e-cigarettes 6 months before pregnancy (27.4%) and 11.3% used e-cigarettes after becoming pregnant. The association between smoking status and use of e-cigarettes before and after becoming pregnant had a small effect size but the mean number of days of e-cigarette use in the past 30 days was higher among current smokers; 3.8 (SD = 4.3) compared with former smokers at 0.3 (0.6), with a large effect size (Cohen’s *d* = 0.96). Most current smokers were highly nicotine dependent (56%), had tried to quit smoking more than four times (26.8%), and had lower confidence in their ability to quit during pregnancy; 4.6 (SD = 2.24) compared with former smokers’ confidence in staying quit during pregnancy at 6.0 (SD = 1.50). Importantly, 71% of women did not receive any smoking cessation treatment during pregnancy. The comparison between groups had a small effect size of 0.44, even though 81% of the former smokers reported not receiving any assistance, compared with 65.9% of current smokers.

This study cohort and mothers who smoked prior to pregnancy in the CDC 2018 Natality dataset had similar distributions for most demographic characteristics. However, our sample had more Black/African Americans (45.2% vs 12.6%) (Cohen’s *d* = 0.96), and more multiple race participants (19.4% vs 4.4%, Cohen’s *d* = 0.92) (see Appendix A). This cohort included more women between 30 and 34 years of age (43.6%) compared with 21.7% in the 2018 Natality data (Cohen’s *d* = 0.56) and more women with a college degree (11.3% vs 4.2%, Cohen’s *d* = 0.59).

## 4. Discussion

Msgs4Moms successfully recruited its full sample size. We recruited a diverse sample that was similar to a national sample of pregnant women with a recent history of smoking. Similar to previous cohorts following smokers or recent quitters during pregnancy and postpartum, most participants enrolled in this cohort had lower education [11,15,18], were facing some kind of financial strain [11,15,18], and were on Medicaid [18]. However, our sample included a greater proportion of African Americans and multiracial participants. This cohort overrepresented Black pregnant women compared with the 2018 Natality data from the CDC. Pregnant women who have a smoking history [45] and are African American [46] are at greater risk of preterm birth, as such, this present cohort represents a group at high-risk of adverse pregnancy outcomes.

Two-thirds of our cohort participants were current smokers and one-third had stopped either in pregnancy or up to 6 months prior. This differs markedly from a cohort assembled 10 years before in the UK in which 57% were current smokers and 43% were recent quitters [23]. Consistent with previously reported socioeconomic disparities in smoking cessation [16], current smokers entering our cohort had less education than recent quitters, fewer were employed and current smokers more often reported financial and housing insecurity. These results were similar to those from Orton et al. that found women who smoke during pregnancy are more likely to hold no educational qualifications, were less likely to own a home, and more likely to engage in unpaid work [23]. Socioeconomic disparities are even more prominent when smokers or recent quitters are compared with nonsmokers [11,14].

More than a quarter of women used e-cigarettes 6 months before pregnancy. E -cigarette use during pregnancy was similar to a national sample of pregnant smokers recruited in 2015 and 2016, that found 17% of pregnant cigarette smokers used e-cigarettes in the past 30 days [47]. In our study, current smokers were using e-cigarettes more often than recent quitters. However, we did not ask participants if they were using e-cigarettes to help them quit smoking. Even though current guidelines do not recommend e-cigarettes as quit assistance [48], pregnant women frequently describe e-cigarettes as safer compared with regular cigarettes and as a quit-smoking resource [47,49].

Most smokers in our study had high levels of nicotine dependence and had tried to quit smoking several times. However, seven out of ten women did not receive any smoking cessation treatment prior to completing our baseline survey. It is possible that they may have received quitting assistance later in pregnancy since most of our sample completed the baseline assessment around 4.5 months pregnant. However, our findings are in line with those of Naughton and colleagues who found that, in early pregnancy, less than half of smokers (43%) reported having talked to a midwife about stopping smoking and fewer had spoken to a general practitioner or nurse (27%). These numbers dropped even more later in pregnancy, with only 27% of smokers reporting speaking to a midwife about stopping smoking [22]. Unfortunately, even though the 5 “A”s (Ask, Advise, Assess, Assist, and Arrange follow-up) has been recommended in many countries as a strategy for health care providers to deliver all the important components of smoking cessation treatment, providers who work with pregnant women rarely address all 5 “A”s [21].

The present cohort has several strengths. First, we used weekly text messages to provide timely data on smoking status, reducing the effects of recall errors. Second, we prospectively collected a range of variables related to tobacco use at five timepoints from pregnancy to 1-year postpartum. Third, we assessed the quality of smoking cessation treatment in close proximity to six health care visits during the study period, rather than relying on retrospective reports at the end of pregnancy. Fourth, this was the first pregnancy cohort to recruit a majority of participants who are African American or Black. A limitation of this research and of our cohort is the fact that smoking status is self-reported. The social stigma of smoking during pregnancy may lead to under-reporting and therefore a response bias. However, other research has shown a high correlation between self-reported smoking and biochemical markers within pregnant populations [50,51]. Additionally, participants were reimbursed for their time responding to the surveys, which can lead to social desirability bias. Another limitation of our study is that we recruited more current smokers than quitters, possibly due to ease of identifying current smokers compared with patients who had already quit smoking in the clinical setting. Last, the small sample size and recruitment from only one region, the US Midwest, may limit the generalizability of the results from this cohort.

When data collection and analysis are complete, this cohort study will extend prior research on smoking during pregnancy and postpartum by employing text message survey delivery to assess smoking at weekly intervals. Frequent assessments of smoking enables us to provide a detailed description of women’s smoking patterns and determinants of both abstinence and smoking. The data collected at several timepoints during the first-year postpartum cover a broad range of smoking-related parameters (including smoking status, cigarettes per day, nicotine dependence, and smoking cessation assistance received during pregnancy), psychosocial (e.g., depression, anxiety, and stress), and socio-demographic characteristics (including but not limited to age, sex, and socioeconomic status). This study presents a novel application of text messages for intensive data collection that can be adapted in low-resource settings. Findings from the cohort study may help to identify potential time-sensitive intervention targets to support smoking cessation during pregnancy and the first year postpartum.

## 5. Conclusions

Msgs4Moms successfully recruited a complete and highly diverse study sample with few differences compared with national data. Baseline data showed that women who continue to smoke during pregnancy are coping with multiple social determinants of health that are well-known variables related to tobacco use. Findings from the cohort study yield insights into both sociodemographic and time-varying variables related to women’s smoking patterns and may help to identify gaps in tobacco treatment and potential time-sensitive intervention targets to support smoking cessation during pregnancy and the first year postpartum.

## Figures and Tables

**Figure 1 ijerph-19-10170-f001:**
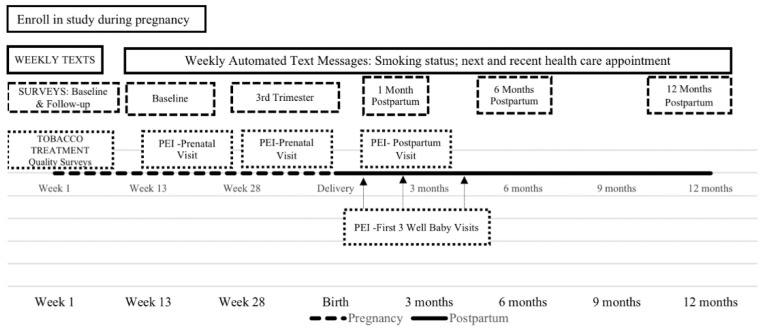
Msg4Moms cohort study flow chart.

**Figure 2 ijerph-19-10170-f002:**
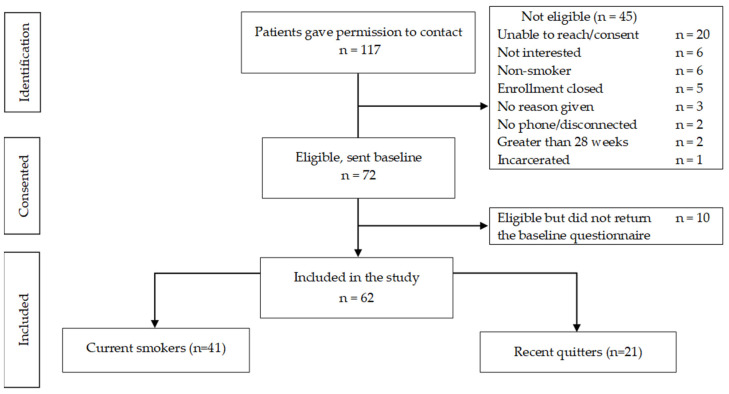
Recruitment flow diagram.

**Table 1 ijerph-19-10170-t001:** Msgs4Moms’s cohort measures and assessment schedule.

Measures	Survey Time Points						
	Baseline	3rd Trimester	1 Month	6 Months	12 Months	Weekly	Following Provider Visits
Socio demographic Factors							
Age	x						
Race and ethnicity	x						
Relationship status	x	x	x	x	x		
Height and pregnancy weight	x	x	x	x	x		
Household income	x						
Employment status	x	x	x	x	x		
Financial burden	x	x	x	x	x		
Housing security	x						
Food security	x						
Baseline pregnancy characteristics							
Weeks gestation	x						
Parity	x						
Wantedness/feelings about pregnancy	x						
Psychological determinants							
Intention to quit/return to smoking	x		x				
Reasons for abstinence	x	x	x	x	x		
Quitting outcome expectations	x	x	x	x	x		
Weight concerns	x	x	x	x	x		
Confidence in quitting	x	x	x	x	x		
Depression, anxiety, and stress	x	x	x	x	x		
Guilt and shame	x						
Health behaviors and other substance use							
Breastfeeding	x		x	x	x		
Physical exercise	x	x	x	x	x		
Marijuana and alcohol	x	x	x	x	x		
Social/environmental influences							
Domestic violence screener	x						
Smoking in social network	x		x	x	x		
Household smoking	x	x	x	x	x		
Home smoking rules	x	x	x	x	x		
Social support	x	x	x	x	x		
Tobacco use and tobacco-related characteristics							
Smoking history/status (6-months prepregnancy)	x						
Current smoking (No. of days smoked, cigarettes per day)	x	x	x	x	x	x	
Nicotine dependence	x	x	x	x	x		
Smoking cessation assistance	x	x	x	x	x		
Craving	x	x	x	x	x	x	
Quitting history	x	x	x	x	x		
Other tobacco use (including e-cigarettes)	x	x	x	x	x		
Tobacco treatment quality							
Provider delivery of the 5 “A”s (Ask, Advice, Assess, Assist, Arrange)							x

**Table 2 ijerph-19-10170-t002:** Baseline demographic and psychosocial characteristics among enrolled pregnant women (n = 62).

Socio-Demographic Characteristics	Total n = 62	Current Smokers n = 41	Recent Quitters n = 21	Cohen’s *d* Values
Age, mean (SD)	28.7 (5.3)	29.2 (5.9)	27.9 (3.8)	0.26
Hispanic/Latina, n (%)	3 (4.8)	0 (0.0)	3 (3.8)	N.A.
Race, n (%)				
Black or African American only	28 (45.2)	21 (51.2)	7 (33.3)	0.41
White only	22 (35.5)	11 (26.8)	11(52.4)	0.61
Multiracial	12 (19.3)	9 (22.0)	3 (14.3)	0.29
Education, n (%)				
Less than high school	11 (17.7)	10 (24.4)	1 (4.8)	1.03
High school/equivalency diploma	22 (35.5)	14 (34.2)	8 (38.1)	0.13
Some college/tech School or higher	29 (46.8)	17 (41.4)	12 (57.1)	0.35
Married/partnered, n (%)	29 (46.7)	17 (41.5)	12 (57.1)	0.35
Health insurance, n (%)				
Medicaid	40 (64.5)	29 (70.7)	11 (52.4)	0.43
Private	19 (30.7)	11 (26.8)	8 (38.1)	0.29
Medicare/other/veterans/uninsured	3 (4.8)	3 (7.3)	0 (0.0)	N.A.
Employment, n (%)				
Employed full-time/part-time/on leave	46 (74.2)	27 (65.9)	19 (90.5)	0.88
Out of work/unable/disabled/other	16 (25.8)	14 (34.1)	2 (9.5)	0.88
No. of weeks pregnant, mean (SD)	18.9 (7.5)	17.8 (7.4)	21.1 (7.5)	0.44
Housing security/living situation, n (%)				
Rent/Other	54 (87.1)	36 (87.8)	18 (85.7)	0.10
Own	8 (12.9)	5 (12.2)	3 (14.3)	0.10
Financial burden/strain, n (%)				
Not enough money to pay bills	31 (50.0)	25 (61.0)	6 (28.6)	0.75
Enough money to pay the bills	31 (50.0)	16 (39.0)	15 (71.4)	0.75
Housing security/not enough money to pay rent, n (%)				
Sometimes/usually/always	31 (50.0)	25 (61.0)	6 (28.6)	0.75
Never/rarely	31 (50.0)	16 (39.0)	15 (71.4)	0.75
Food security/not enough money to buy meals, n (%)				
Sometimes/usually/always	42 (67.7)	28 (68.3)	14 (66.7)	0.04
Never/rarely	20 (32.3)	13 (31.7)	7 (33.3)	0.04
Previous pregnancy	42 (67.7)	29 (70.7)	13 (61.9)	0.22

**Table 3 ijerph-19-10170-t003:** Tobacco-related characteristics among pregnant women from obstetric and family practice clinics in US Midwest.

Variables	Total n = 62	Current Smokers n = 41	Recent Quitters n = 21	Cohen’s *d* Values
Tobacco use characteristics				
Past 6 month tobacco use/prepregnancy, n (%)				
Every day	54 (87.1)	38 (92.7)	16 (76.2)	0.76
Some days	8 (12.9)	3 (7.3)	5 (23.8)	
No. of cig. per day 6-months prepregnancy, mean (SD)	10.56 (7.1)	11.7 (7.4)	8.4 (6.1)	0.46
Cigarettes use in the past 30 days, n (%)	41 (66.1)	41 (100.0)	N.A.	N.A.
No. of days of tobacco use in the past 30 days, mean (SD)	12.6 (13.1)	19.1 (11.6)	N.A.	N.A.
No. cigarettes/day in the past 30 days, mean (SD)	N.A.	6.5 (5.7)	N.A.	N.A.
Use of other tobacco product during pregnancy, n (%)	12 (19.3)	10 (24.4)	2 (9.5)	0.62
E-cigarette use 6-months prepregnancy, n (%)	17 (27.4)	12 (29.3)	5 (23.8)	0.15
E-cigarette use after becoming pregnant, n (%)	7 (11.3)	4 (9.8)	3 (14.3)	0.24
No. of days of e-cigarettes use in 30 days, mean (SD)	2.3 (3.5)	3.8 (4.3)	0.3 (0.6)	0.96
E-cigarette refills/cartridges daily, mean (SD)	0.9 (0.7)	1.0 (0.0)	0.7 (1.1)	0.41
Nicotine dependence, n (%)				
Smoke within 30 min after waking up	N.A.	23 (56.1)	N.A.	N.A.
Cigarettes craving (0–5), mean (SD)	1.8 (1.6)	2.4 (1.5)	0.7 (1.2)	1.08
Quitting attempt, ≥day, n (%)	N.A.	25 (60.1)	N.A.	N.A.
Once	N.A.	7 (17.1)	N.A.	N.A.
2–3 times	N.A.	7 (17.1)	N.A.	N.A.
4 or more	N.A.	11 (26.8)	N.A.	N.A.
Behavioral and cognitive factors				
Intention to quit/Likely to smoke				
At the end of pregnancy (1–5), mean (SD)	1.8 (1.2)	2.0 (1.3)	1.6 (1.2)	0.29
6 months after the baby is born (1–5), mean (SD)	2.1 (1.2)	2.2 (1.3)	2.0 (1.1)	0.24
Confidence in quitting during pregnancy (1–7), mean (SD)	5.11 (2.1)	4.6 (2.2)	6.0 (1.5)	0.67
Smoking cessation assistance during current pregnancy, n (%)				
Booklets, videos, or other materials	3 (4.8)	3 (7.3)	0 (0.0)	N.A.
A quit smoking program	3 (4.8)	3 (7.3)	0 (0.0)	N.A.
Medication	5 (8.1)	4 (9.8)	1 (4.8)	0.43
Other	9 (14.5)	6 (14.6)	3 (14.3)	0.02
None	44 (71.0)	27 (65.9)	17 (81.0)	0.44

## Data Availability

The data used in this study are available on request from the corresponding author.

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
