# Peer review of "Quitting Smoking before and after Pregnancy: Study Methods and Baseline Data from a Prospective Cohort Study"

_ijerph, 2022, doi:10.3390/ijerph191610170_

Round 1

Reviewer 1 Report

This paper represents an interesting dual-purpose study highlighting the prospective protocol and presenting the preliminary results from the baseline sample. It is well-written and certainly of high interest and importance for researchers and clinicians. Just a few notes:

Abstract: Perhaps add something about the baseline data presentation. As it stands now it sounds like this will be a protocol paper but then there is baseline data presented. 

Introduction: Is there more recent population data from the US? All but 2 of the references are from prior to 2020. 

Methods: Is the eligibility criteria of "at least 100 cigarettes" lifetime cigarettes?

Will the retention efforts be added as covariates to the final analysis? Perhaps those who had to complete the study by phone or email might show different results from those who complete via electronic questionnaire

Is there more recent Natality data from the CDC? What is quoted here is from 2018 but it seems like there is more recent data available. Consider updating this section.

Results: The results for current smokers vs. those who have recently quit is compelling. How do these demographics compare to never smokers? It might make for a compelling comparison if you don't find large differences between smokers and recent-quitters but can say that both differ significantly from non-smokers.

Overall, this will be a very impressive study and I look forward to the pleasure of reading the results!

Author Response

July 28, 2022

Ref.: (ID ijerph-1728271) “Quitting smoking before and after pregnancy: study methods and baseline data from a prospective cohort study”.

Dear Ms. Veronica You and the reviewer,

Please find our revised manuscript attached. We appreciate the reviewers’ careful evaluation of our paper and their constructive comments. We have included the responses to all three reviewers’ concerns below. We appreciate their interest in the article very much. 

Corresponding changes in the manuscript are highlighted as track changes. We have also supplied a clean version of the manuscript with changes saved. Please let us know if there is anything else we can do to aid in the review of the paper. 

Best regards,

Erica Cruvinel

Reviewer 2 Report

This study focuses on a stigmatised topic: this fact, in addition of the fact that partecipats recived an incentive, may have influenced the behaviour of respond to surveys.

Author Response

(The authors gave the same response as above.)

Reviewer 3 Report

The aim of this study according to the authors was to report methods used to assemble a prospective cohort for investigating smoking patterns, determinants of smoking and abstinence, and tobacco treatment quality across pregnancy through one-year postpartum.

The study focuses on a population of special interest. Studying smoking behavior in pregnant women is especially relevant because of the implications smoking has on the woman and the baby. But the present manuscript has serious limitations that make it difficult to publish.

The main limitation is that this manuscript does not contribute significantly to the scientific literature. The authors provide information about the characteristics of the sample and compare their data with the cohort sample to the 2018 Natality data from CDC. As the authors note, generalized estimating equations (GEE) could provide more relevant information to describe  women’s patterns of smoking behavior and their associations with psychosocial variables.

Other aspects to point out more specifically:

-        The introduction section is unclear. The authors point out information on some variables to be analyzed in the study, but it is confusing. A better description of the variables to be analyzed is required.

-        The authors should clarify what the objective of the study is. Due to the large number of variables analyzed, it may be difficult to understand. The objectives stated in the abstract, main manuscript and even in the title must be similar and specific.

-        I would recommend that the informed consent be in writing, not verbal as indicated by the authors.  

-        Recent smokers quitters group is significantly smaller than current smokers group. 

-        Some of the variables analyzed have many categories. This limits the statistical analysis. I would recommend grouping the categories or increasing the number of participants.

-        The authors use terms that are inappropriate for the type of analysis conducted (i.e., moderators and mediators; predictors)

-        The results section is scarce. The authors propose to evaluate many variables but provide limited information on the results obtained. Other types of statistical analysis such as logistic regression could be performed.

-        Information on the possible contributions of this study should be improved.

Author Response

(The authors gave the same response as above.)

Round 2

Reviewer 3 Report

Please find the attached file with detailed comments.

Author Response

August 7, 2022

Ref.: (ID ijerph-1728271) “Quitting smoking before and after pregnancy: study methods and baseline data from a prospective cohort study”.

Dear Ms. Veronica You and the reviewer,

Please find our revised manuscript attached. We appreciate the reviewer's constructive comments. We have included the responses to the reviewers’ concerns below.

Corresponding changes in the manuscript are highlighted as track changes. We have also supplied a clean version of the manuscript with changes saved. Please let us know if there is anything else we can do to aid in the review of the paper. 

Best regards,

Erica Cruvinel
